# Looking beyond Typical Treatments for Atypical Mycobacteria

**DOI:** 10.3390/antibiotics9010018

**Published:** 2020-01-03

**Authors:** Clara M. Bento, Maria Salomé Gomes, Tânia Silva

**Affiliations:** 1i3S—Instituto de Investigação e Inovação em Saúde, Universidade do Porto, 4200-135 Porto, Portugal; clara.bento@i3s.up.pt (C.M.B.); tania.silva@ibmc.up.pt (T.S.); 2IBMC—Instituto de Biologia Molecular e Celular, Universidade do Porto, 4200-135 Porto, Portugal; 3ICBAS—Instituto de Ciências Biomédicas Abel Salazar, Universidade do Porto, 4050-313 Porto, Portugal

**Keywords:** nontuberculous mycobacteria, antibiotics, ionic liquids, iron chelators, antimicrobial peptides, bacteriophages, host-directed therapies

## Abstract

The genus *Mycobacterium* comprises not only the deadliest of bacterial pathogens, *Mycobacterium tuberculosis*, but several other pathogenic species, including *M. avium* and *M. abscessus*. The incidence of infections caused by atypical or nontuberculous mycobacteria (NTM) has been steadily increasing, and is associated with a panoply of diseases, including pulmonary, soft-tissue, or disseminated infections. The treatment for NTM disease is particularly challenging, due to its long duration, to variability in bacterial susceptibility profiles, and to the lack of evidence-based guidelines. Treatment usually consists of a combination of at least three drugs taken from months to years, often leading to severe secondary effects and a high chance of relapse. Therefore, new treatment approaches are clearly needed. In this review, we identify the main limitations of current treatments and discuss different alternatives that have been put forward in recent years, with an emphasis on less conventional therapeutics, such as antimicrobial peptides, bacteriophages, iron chelators, or host-directed therapies. We also review new forms of the use of old drugs, including the repurposing of non-antibacterial molecules and the incorporation of antimicrobials into ionic liquids. We aim to stimulate advancements in testing these therapies in relevant models, in order to provide clinicians and patients with useful new tools with which to treat these devastating diseases.

## 1. NTM Infections: Epidemiology and Clinical Presentations

### 1.1. Epidemiology

Nontuberculous mycobacteria (NTM) are bacterial species that fall within the *Mycobacterium* genus, but which are outside the *M. tuberculosis* complex or the species *M. leprae* [1,2]. Around 190 NTM species are described, which are classified into two groups, i.e., the slowly growing (SGM), and the rapidly growing mycobacteria (RGM), according to the time required to form visible colonies in solid media (more or less than seven days, respectively). The most relevant NTM species for human disease are the members of the *M. avium* complex, *M. kansasii* and *M. xenopi* (all SGM), *M. abscessus* complex, *M. chelonae*, and *M. fortuitum* (the last three are RGM) [3,4,5]. All of these are environmental organisms, present mainly in soil and water. However, the incidence of human infections by NTM is increasing significantly worldwide. Although some mycobacterial species may cause other forms of disease, such as cutaneous infections, in this review, we will focus mainly on pulmonary disease. The exact numbers are difficult to find, as in most countries, the reporting of infections by NTM is not mandatory [5,6,7]. According to the available data, the incidence of disease varies considerably with NTM species, geographic distribution, sex, race/ethnicity, age, and risk factors (e.g., concomitant debilitating diseases). Women are at higher risk of infection, alongside people with Asian ancestry and from the Southern United States, such as Hawaii [8,9]. Species of the *M. avium* complex (MAC) are the most common causes of NTM infections and are mainly responsible for the observed increase in disease incidence [3,5,6,10]. Their resistance to antibiotics is growing; therefore, the treatment used today is a multidrug therapy comprising at least three antibiotics, with treatments taking from six months to years. However, a very long multidrug regimen like this results in several issues for patients, thereby decreasing the probability of success of the treatment. It is therefore urgent to find a new strategy to treat mycobacterial infections.

The fact that NTM with highly-hydrophobic cell walls, which facilitates aerosolization and surface adherence, are widely distributed in the environment, may explain their highly-infectious behavior. Moreover, NTM are able to survive in harsh environments, being exquisitely resistant to chlorine-based disinfectants, and their capacity to adhere to surfaces and form biofilms allows them to persist for long periods of time [10,11]. Biofilm formation and intercellular communication by quorum-sensing provide a high level of resistance to unfavorable environments and to the action of disinfectants and antibiotics. NTM, especially RGM, are also known to adhere to biomaterials, creating biofilms in medical devices, such as catheters, which may cause pathologies which are difficult to diagnose and treat [12].

### 1.2. Relationship between Tuberculosis and NTM Infections

Curiously, it has been reported that a local decline in tuberculosis (TB) incidence is coincidental with an increase in infections caused by NTM [5,13,14]. There is no single explanation for this phenomenon, but some can be hypothesized, e.g., cases of cross-immunity between *M. tuberculosis* (Mtb) and NTM, in which each type of mycobacteria sensitizes the host to a second exposure of the other [15]. Also, better public health conditions can be, in this case, a double-edged sword. While improved ventilation and plumbing were essential to reducing TB incidence, centralized water supply systems, the disinfection of drinking water, and the habit of showering instead of tub-bathing are associated with NTM colonization, leading to the selection of these microorganisms due to their resistance to chlorination and higher exposure to mycobacteria through aerosolization [10,11,16,17]. Indeed, *Mycobacterium* was the most prevalent genus detected in showerheads throughout Europe and the United States, with a higher incidence in showerheads receiving municipal water (chlorine-treated water) [11,16]. Most surprisingly, Gebert et al. found that regions in the United States with high levels of NTM lung disease overlapped with high abundances of potential pathogenic NTM species detected in showerheads [16]. In health-care centers, the prevalence of NTM in plumbing systems is also very high, resulting in contamination and outbreaks associated with exposure to NTM-contaminated tap water of wounds, surgical instruments, prostheses, and dialysis-related equipment, among others [4,7]. Another important factor is the misdiagnosis of TB. In developing countries, where the incidence of TB is high, diagnoses usually do not distinguish between NTM and TB, only detecting a pulmonary mycobacteriosis. As a result, a significant number of NTM infections are classified as TB, and, as a consequence, these patients receive anti-TB therapy that is not effective against NTM. Thus, these patients are then classified as having either chronic or multidrug resistant (MDR) TB, with a high impact not only on their health status, but also on the associated health-costs [10].

It must always be borne in mind that higher clinical awareness and better diagnoses may explain the increased incidence rate of NTM infections in recent years. However, the increasing number of studies published worldwide every year reporting cases of NTM infection and the accumulated evidence they present indicate that these infections are indeed on the rise. Alterations in human lifestyle, including the massive use of plumbing water systems, the increased life expectancy, the growing use of immunosuppressive therapies, and increased human mobility and trade, all may be contributing to this trend [10].

### 1.3. Clinical Presentations

NTM human disease is thought to be acquired only from environmental sources, like exposure to tap water, soil, or the use of showers or cooling/heating devices, with no definitive evidence of human-to-human transmission [7,10,11,16,17,18]. However, there have been recent reports of possible transmission of *M. abscessus* between cystic fibrosis patients [3,19]. NTM disease can manifest essentially in four clinical syndromes: pulmonary disease (the most common form), skin and soft tissue infections, lymphatic disease, and disseminated disease [4]. Pulmonary disease usually occurs in individuals that are sensitized by other lung-associated diseases, like bronchiectasis, chronic obstructive pulmonary disease, cystic fibrosis, pneumoconiosis, or prior TB, among others, or with a compromised immune system due to conditions like rheumatoid arthritis, HIV-infection, cancer, or organ transplant [3]. Skin and soft tissue infection is associated with the contamination of wounds or prostheses and surgical procedures, and is more commonly associated with infection with RGM, like *M. abscessus* complex, *M. chelonae*, and *M. fortuitum*. The disseminated disease occurs in individuals with severely-compromised immune systems, especially in patients infected with HIV. NTM lymphatic disease is more commonly seen in immunocompetent children under 5 years old [4,20].

## 2. NTM Biology and Interaction with the Host Cell

Mycobacteria are aerobic bacilli with a characteristic dense and complex cell wall. A thick layer of peptidoglycan gives structural strength to this wall. Additionally, mycolic acids, i.e., complex fatty acids of 60 to 90 carbons, are esterified with arabinogalactan, a sugar polymer, and confer high hydrophobicity and impermeability. The exterior part of the wall is composed of glycolipids and lipoglycans that interact with the mycolic acids and are covered by loose proteins, lipids, and glycans [21]. All these lipid layers make mycobacteria acid-fast, so they are not stained by the Gram method; instead, they require a harsh procedure like the Ziehl-Neelsen staining to be visualized [22].

Mycobacteria enter the body either by the respiratory (via aerosols) or gastrointestinal tract, or by wound and prosthesis contamination. Inside the host, these bacteria are found inside different cell types, but their main and most studied host cell is the macrophage. When mycobacteria infect macrophages, they localize in tight individual vacuoles, where they are able to grow exponentially [23]. It was observed that the vacuole also divides, accompanying the multiplication of the mycobacteria in order to maintain them in tight compartments. This strategy allows the mycobacteria to better control the fusion of their vacuoles with other vesicles that are present in the cell, like lysosomes. The arrest of the maturation of early mycobacterium-containing phagosomes is a well-known protective mechanism of these pathogens [23,24,25]. This mechanism is believed to prevent the exposure of the mycobacteria to the lysosome acidic environment, hydrolytic enzymes, and macrophage antigen-presenting organelles [26]. Additionally, it may facilitate the access of mycobacteria to nutrients, like iron, located in cell membrane-derived vesicles with which the early mycobacterium-containing phagosomes are able to interact [23,27]. Macrophages may also inhibit bacterial growth by mechanisms that involve oxidative damage, e.g., by nitric oxide, hydrogen peroxide, and superoxide. Interestingly, the NTM *M. avium* was shown to be more resistant than Mtb to both phagosome-lysosome fusion and oxidative damage [23]. This may be explained by a better adaptation of *M. avium* to harsh natural environments [27], in contrast to Mtb, which can only survive inside the human host. On the other hand, this paradox indicates that macrophages must have other antimicrobial mechanisms that can control and eliminate NTM in immunocompetent individuals [28]. An alternative way of restricting the growth of intracellular pathogens is by depriving them of essential nutrients. Macrophage activation may prevent the interaction of the *M. avium*-containing phagosomes with endosomes that carry nutrients [27,29]. On the other hand, the protein SLC11A1 may contribute to mycobacterial growth restriction by pumping iron and other cations out of the pathogen-containing phagosome [28]. Macrophage autophagy and apoptosis have been increasingly identified as important mechanisms in the control of intracellular pathogens. However, it was shown that MAC are able to escape apoptotic bodies to the extracellular space, infecting and spreading to healthy cells and tissues [30].

The control of mycobacterial infection by the host relies on an immune response centered in CD4^+^ T cells that produce the macrophage-activating cytokine IFN-γ. The depletion of IFN-γ or of upstream cytokines such as IL-12 and IL-18, which are necessary for the maturation of CD4^+^ T cells into IFN-γ-producing Th1 cells, exacerbates mycobacterial infection [27,31,32,33]. TNF is another important macrophage-activating cytokine. It increases the macrophage’s antimycobacterial activity in vitro. Additionally, in vivo, TNF is involved in maintaining the structure and integrity of granulomas, i.e., the inflammatory lesions that are a hallmark of mycobacterial infections [27].

## 3. Current Treatments Available for NTM Infections

### 3.1. Limitations and Challenges

Some of the clinical features associated with pulmonary NTM infection are similar to those of TB, caused by the closely-related pathogen, Mtb [34]. It is therefore comprehensible that the first treatments used for NTM infections were antituberculous drugs, which were successful in some cases, but demonstrated lower activity in this setting than against Mtb [35]. With the increasing incidence of NTM infections, mainly of MAC species, and the emergence of AIDS, a new strategy had to be adopted. The discovery of antibiotics that have a better effect against NTM infections than the antituberculous drugs which were previously used, such as the macrolides, revolutionized the treatment of NTM lung disease [35]. However, progress has been slow, with some renewed interest in recent years. The high number of NTM species and the similarity of the clinical features presented, concomitant with differences in the susceptibility to the available antibiotics, often hampers correct diagnoses and treatment. One of the important challenges when choosing an efficient treatment for NTM infection has been the lack of correlation between in vitro susceptibility patterns and the clinical response. This has contributed to delays in the implementation of appropriate guidelines, and has prompted treatment failure and the development of resistance [4]. For most NTM species, there are no evidence-based treatment recommendations, and clinicians are left to make decisions on a case-by-case basis.

In 2007, the American Thoracic Society (ATS) and the Infectious Diseases Society of America (IDSA) [4], and more recently the British Thoracic Society (BTS) [36], released a series of guidelines for the treatment of NTM pulmonary infections. The main consensus recommendation is the use of a macrolide-based multidrug regimen to treat pulmonary disease caused by MAC or *M. abscessus* complex (MABC) until the patient is sputum culture-negative for one year, which results in very long (months to years) treatments. Given the high toxicity and relatively low efficacy of available drugs, health professionals sometimes opt for no treatment in less severe cases, and patients are kept under clinical observation only. The prescription of multidrug therapy is fundamental to avoid macrolide resistance and to prevent unnecessary deaths in more severe cases, since surgical lung resection is often the only solution for patients who fail drug therapy due to antibiotic resistance [37]. Cases of re-infection with MAC after or during therapy are also of concern, affecting mainly patients with a severely immunocompromised system [35].

At the core of the failure of NTM infection treatments may be the lack of adherence to the established guidelines by health professionals [38,39,40,41,42]. Adjemian et al. performed a US survey where clinicians were asked to report treatment choices for NTM-infected patients treated in 2011 [39]. Surprisingly, of the 744 treatments prescribed against MAC, only 13% met the ATS/IDSA guidelines, 56% did not include a macrolide, and 16% were on a macrolide monotherapy. In the case of the 174 MABC infections, 64% of them were not treated with a macrolide [39].

### 3.2. Base-Line Treatments for NTM

The basis of all current treatments for NTM infections is macrolides. These antibiotics have the best correlation between in vitro susceptibility results and clinical (in vivo) response [35,37]. Clarithromycin or azithromycin are the usual options, with no significant differences in response between the two [43]. Clarithromycin has been more extensively studied, and it is often the preferred choice. However, it interferes with cytochrome P 3A enzymes, which may result in undesirable interactions with other drugs. To avoid this effect, azithromycin may be a better choice [44]. Macrolides inhibit protein synthesis in bacteria by binding to the 50S ribosomal subunit and preventing the elongation of the nascent peptide chain [44,45]. Resistance to macrolides in several RGM species, such as *M. abscessus*, as well as in Mtb, are related to the inducible macrolide resistance gene, *erm*. The activation of this gene reduces the binding of macrolides to the ribosome by the methylation of an adenine in the 23S rRNA [37,43,44,46]. A regimen of monotherapy with macrolides is, thus, very dangerous, as it will often lead to drug resistance and consequent treatment failure, associated with increased levels of mortality. The recommended treatment for MAC infection is a three-drug macrolide-based regimen with ethambutol and a rifamycin [4,36,43]. Since, as stated above, for many other species of NTM, there are no consistently effective drug combinations, therapy is often based on this same, three-drug macrolide-based regimen, eventually incorporating the results of the in vitro susceptibilities of the particular clinical isolates.

Ethambutol interferes with mycobacterial cell wall synthesis by the inhibition of arabinosyl transferases, which affects the synthesis of arabinogalactan and lipoarabinomannan. Its ability to alter the permeability of the cell wall, allowing the passage of other antimycobacterial drugs to occur, is its major advantage. Besides that, there are no known negative drug interactions with ethambutol. This antibiotic has good activity against SGM, including MAC, but high levels of resistance in RGM species exclude its use in this case [37,44].

Rifamycins like rifampicin and rifabutin complete the bases of the treatment for NTM. They bind and inhibit DNA-dependent RNA polymerases, interrupting RNA synthesis early in the transcription process, displaying bactericidal activity [44]. They are a crucial part of the treatment against both NTM and Mtb, as they are able to retain bactericidal activity against intramacrophagic and nonreplicating bacteria [47], as well as to sterilize the granulomas’ necrotic center [48]. Co-infection with HIV may decrease the absorption of rifamycins, and notably, rifampicin is known to reduce the concentrations of several antiretroviral drugs. Rifamycins are involved in multiple interactions with other drugs, since they are strong inducers of CYP3A4, among other enzymes, and studies with patients infected with NTM showed low tolerance to rifabutin, although it was shown to have better in vitro and in vivo activity against MAC than rifampicin [37,43,44,49]. From the three types of drugs recommended to treat MAC infections, rifamycins contribute the most to suboptimal pharmacokinetics and pharmacodynamics parameters. In fact, poor antimycobacterial activities and several interactions with other agents, resulting in the reduction of the plasma concentration of these drugs, require the increase of dosages to levels that may become intolerable to patients [43,46]. In vitro susceptibilities to rifamycins and ethambutol do not correlate with treatment response, and thus, these results do not help clinicians to make treatment decisions. Rifamycins are a crucial part of the base treatment for MAC and *M. kansasii*; however, they are not recommended for the treatment of MABC, since these species are resistant to rifampicin [50]. Interestingly, in 2017 a screening of thousands of FDA-approved drugs showed that rifabutin is bactericidal against all MABC species, suggesting that this drug may be repurposed for the treatment of *M. abscessus* infections, and could also help to develop more potent ryfamicins [50,51].

### 3.3. Second-Line Treatments for NTM

Second-line drugs are of extreme importance in specific cases, like severely-disseminated or recalcitrant disease, and are essential in cases of macrolide-resistant MAC. Aminoglycosides, like amikacin or streptomycin, are protein synthesis inhibitors and are most often used in such cases [4]. It has been shown that in vitro minimal inhibitory concentration (MIC) for amikacin correlates well with clinical response, helping to predict treatment success [52]. Amikacin liposome inhalation suspension (ALIS) is a new promising alternative for these types of infections, and is, at present, in phase III trials for recalcitrant MAC lung disease after encouraging results in phase II [43,53].

Fluoroquinolones, like moxifloxacin, are direct inhibitors of bacterial DNA synthesis, and have been included in the therapy of lung disease caused by macrolide-resistant MAC, although there is little evidence to support this. Moreover, their use as first-line drugs, as well as in a macrolide/fluoroquinolone regimen, is inadvisable due to the risk of drug resistance and cardiac toxicity [36,37,43,44,46]. Nonetheless, fluoroquinolones have significant activity against *M. kansasii*, and are of extreme importance in cases of rifamycin-resistant *M. kansasii* [54].

Clofazimine is a riminophenazine, acting as a prodrug that releases reactive oxygen species. It is part of the multidrug standard treatment used against *M. leprae*, and more recently, it has been repurposed for MDR-TB [55,56]. In the case of NTM, it has shown favorable activity against pulmonary disease [57,58], and some data suggest that low tolerance to rifamycins can be overcome with clofazimine combined with a macrolide and ethambutol [59,60]. Clofazimine-containing regimens plus amikacin, isepamicin, or bedaquiline seem promising [54,61,62,63,64]. Moreover, due to its high penetration in skin and soft tissue, it has potential applications against NTM skin and soft tissue disease, like those caused by RGM [54].

Linezolid was the first member of the class of oxazolidinone antibiotics to be approved by the FDA to combat Gram-positive infections [65]. It has also been used as a second-line drug for the treatment of MDR-TB [55] and in patients who fail a macrolide-based regimen [4]. However, its poor tolerability and high toxicities in long-term treatment discourage its use [66]. Nevertheless, there are some data demonstrating that linezolid and other oxazolidinones [67,68,69,70] have potential in the treatment of NTM infections [68,71], especially those caused by RGM [72,73]. Several case reports describe the use of linezolid as being successful in the treatment of skin and soft tissue diseases caused by *M. chelonae* and *M. abscessus* [74,75,76,77,78].

Bedaquiline is a second-line antibiotic used in cases of MDR-TB [79,80]. It belongs to a new class of drugs approved in 2012 called diarylquinolines that act by binding and inhibiting the mycobacterial ATP synthase, resulting in ATP depletion and cell death [81]. Bedaquiline has been shown to have promising in vitro [82,83,84,85,86,87,88,89,90] and in vivo [91,92,93] activities against several species of NTM, including SGM and RGM. Moreover, preliminary results have shown that bedaquiline has potential clinical applications in patients with MAC or MABC lung disease [94,95]. The combination of bedaquiline with clofazimine is a promising addition to the NTM therapy [63]; however, there is emerging evidence that it has an antagonistic effect with clarithromycin and β-lactams [96], and that rifamycins reduce bedaquiline concentrations [97].

Delamanid belongs to a novel class of antibiotics, cyclic nitroimidazole, approved in 2014, to be used against MDR-TB. It acts as a prodrug, inhibiting the synthesis of mycolic acids [98]. Little data has been published on the activity of delamanid against NTM species, and what does exist has shown inconsistent results [98]. However, some studies have reported antimycobacterial activity, in particular against SGM species [83,90,99].

### 3.4. The Special Case of M. abscessus

Diseases caused by species from the MABC are extremely difficult to cure, being, in some cases, comparable to MDR-TB. Macrolides are the only drugs with proven efficacy against MABC; however, as most clinical isolates have an active *erm* gene, it is of utmost importance to first determine which subspecies is the disease’s causative agent. No evidence-based treatment exists, and due to its intrinsic resistance to many of the available drugs (e.g., all first-line anti-TB drugs), treatment is often aggressive, requiring a combination of a macrolide (when suitable) with at least two parenteral drugs, according to in vitro susceptibilities, like amikacin, linezolid, or β-lactams (imipenem or cefoxitin) [54]. However, it must be kept in mind that for most drugs, in vitro susceptibility does not correlate with clinical outcomes, leaving surgical resection as the last option for these patients. Recently, β-lactams (inhibitors of peptidoglycan synthesis), especially dual combinations of these drugs, have been repurposed for the treatment of MABC lung disease, with favorable in vitro [100,101] and in vivo [102] results. Moreover, the combination with β-lactamase inhibitors (i.e., avibactam) improved the activity of several carbapenems and cephalosporins [100,101,102,103,104,105,106,107,108].

### 3.5. New Antimycobacterial Compounds in Preclinical Studies

Besides the discovery and development of new antibiotics, several reports have described the antimycobacterial activity of other types of molecules which are currently in preclinical studies. In Table 1, we summarize the most recent studies.

## 4. Alternative Approaches—Beyond Typical Treatments

In light of the many difficulties and limitations identified above, it is evident that new antimycobacterial strategies must be developed to achieve better global control of NTM infections. New, alternative treatments should be able to allow shorter treatment durations, reduced daily pill burden and dose frequency, treatment of multidrug resistant strains, and the possibility of co-administration with other relevant drugs (e.g., anti-HIV drugs). In recent years, renewed efforts have been made towards the discovery of alternative approaches to tackle NTM infections. In the next sections, we will cover some of the most promising, new alternatives reported in the literature, which include the repurposing of conventional drugs and the use of ionic liquids, antimicrobial peptides, bacteriophages, iron chelators, and host-directed therapies (Figure 1).

### 4.1. Repurposing Old Drugs

The discovery of new drugs is extremely slow and costly. Repurposing drugs previously validated for other diseases accelerates the process while avoiding many of its difficulties. With this in mind, in recent decades, many researchers have turned to screening hundreds to thousands of previously developed or approved drugs for possible activity against different pathogens, including NTM [50,126,127]. These screenings can reveal a number of interesting hits that can be further explored in terms of in vitro and in vivo antimicrobial activity, mechanism of action, toxicity, resistance, synergism with other available drugs, and the possibility (or need) of fine-tuning to meet the new purpose. In Table 2, we list some of the promising compounds that could be repurposed for the treatment of NTM infections.

In this list, we find very different molecules, either of natural origin, such as carvacrol, or that have been originally developed for unrelated diseases, such as the antipsychotic, thioridazine. Additionally, several of these drugs were previously shown to have antiparasitic activity, including antimalarial agents. Chloroquine (CQ) was first synthesized during World War II, being identified as the most promising antimalarial drug due to its good efficacy, low toxicity, tolerable adverse effects, and affordability [141,142]. Besides malaria, it was demonstrated that CQ has anti-HIV-1 and anti- *M. avium* activity in vitro [139], suggesting it can be viewed as a multiversed drug in the treatment of AIDS-related opportunistic infections. Experiments in our laboratory showed that CQ has a significant inhibitory effect in vitro against *M. avium* [139]. That inhibitory effect was also evident in vivo: BALB/c and C.D 2 mice infected with *M. avium* 2447 SmT treated with 30 mg/kg of CQ every other day showed a significant decrease of bacterial loads in the liver (unpublished results). Primaquine (PQ) is the most effective and least toxic 8-aminoquinoline to have been used as an antimalarial since the 1950s [143]. More recently, it was reported that primaquine at 5 µM was able to inhibit the intracellular growth of Mtb [144], and some PQ-derivatives tested against Mtb, *M. paratuberculosis*, and MAC showed strong antimycobacterial activity [140].

### 4.2. Ionic Liquids

In the rescue of old and less utilized drugs, unfavorable pharmacological properties, such as low solubility, spontaneous crystallization, and the high dosage needed to achieve the desirable effects or toxicity to the host infected cells, are common problems. These difficulties can be overcome by the synthesis of noncrystalline forms of those drugs, i.e., ionic liquids (ILs), which are organic salts made by the combination of the active pharmaceutical ingredient in its cationic or anionic form and an inert counterion, or a counterion which is of additional biological interest. The cost of this synthesis can even be lowered by, for example, combining existing drugs of opposed polarities [145,146,147].

With remarkable physical and chemical properties, ILs were first used to improve the performance and safety of chemical procedures as green-solvents. Recent studies regarding the interaction between ILs and biomaterials have revealed their strong potential to improve sensors and drug delivery systems [147]. It has been demonstrated that ILs work well as antimicrobial agents, affecting Gram-positive and -negative bacteria, but also mycobacteria and fungi [147]. The right combination of cations and anions can provide innovative compounds that help combat resistance issues. The mechanism of action of ILs, as compared to conventional drugs, is not yet fully understood. However, structural characteristics such as the length of the cation side chain or the presence of polar functional groups can alter properties, such as lipophilicity and surface tension, that are known to influence the activity of the compounds [147]. As active pharmaceutical ingredients, ILs are emerging as a promising means by which to overcome issues related to polymorphism while improving solubility and bioavailability in a cost-effective way.

The combination of anionic ampicillin with organic cations resulted in ILs with activity against Gram-negative bacteria resistant to antibiotics [148]. ILs derived from a classical antimalarial drug, primaquine, an 8-aminoquinoline, were found to exhibit improved in vitro performance in comparison to primaquine, and better in vitro activities than their covalent analogs [145]. ILs derived from N-cinnamoylated CQ conjugates were reported to have similar activity against *Pneumocystis jirovecii* than their covalent equivalents; however, they were shown to be less cytotoxic to two different cell lines than their covalent equivalents [149]. Recently, our group tested ILs based on the cationic molecule of CQ and differently-substituted anionic cinnamoyl groups against *M. avium*. Although these ILs did not show a better inhibitory effect than their covalent equivalents, they were significantly more soluble and less toxic to macrophages harboring the bacteria [150]. These results confirm the ability of ILs to overcome the pharmacological issues associated with their drugs of origin.

ILs are very promising in terms of efficacy, affordability, and as a form of bypassing problems of resistance. Their properties allow conventional drugs to be combined, with different effects but with a joint purpose. For example, bacterial infection in an AIDS patient could be treated in the future with only one medicine, an IL comprising an antiretroviral ion and a counterion with antibacterial activity.

### 4.3. Antimicrobial Peptides

Antimicrobial peptides (AMPs) compromise a vast array of small peptidic compounds that exhibit antimicrobial activity against a wide variety of pathogens (viruses, bacteria, fungi, and protozoans). AMPs are widespread in nature, i.e., they are present in almost all living organisms as part of their innate immune mechanisms. These compounds are highly diverse in their length, sequence, structure, source, and activity; as a result of such diversity, there is no universal target or defined mechanism of action. Nonetheless, it is well known that they interact with pathogens’ cytoplasmic membranes, disrupting them or not, which can lead to cellular death; they can also act intracellularly, interacting with fundamental molecules, such as DNA, RNA, or proteins [151]. Moreover, AMPs can also have immunomodulatory properties, and for that reason, they have been called “host defense peptides” [152,153]. This multitude of actions increases AMP efficacy, but above all, enables them to escape potential microbial resistance mechanisms. Of interest, AMPs do not distinguish between metabolically-active or inactive microbial cells, as opposed to conventional antibiotics [154], which increases their scope of action, especially against biofilms. Bearing all this in mind, AMPs are an attractive alternative approach in the fight against bacterial diseases. AMPs have been described to be active against mycobacteria, both through directly killing or immunomodulation (reviewed in [153,155,156]). Most notable is the role of the human peptide cathelicidin LL-37, which induces autophagy and phagosomal maturation in mycobacteria-infected macrophages via the activation of vitamin D signaling pathways [157]. Moreover, it was shown that the production of cathelicidins and several defensins (e.g., human neutrophil peptide (HNP) and beta-defensins) is upregulated during mycobacterial infections [155]. Additionally, in some cases, the effect of AMPs in vivo is achieved at lower concentrations than those necessary for in vitro antimicrobial activity, indicating that their mechanism of action is much more complex than just the direct membrane disruption [158,159]. The combination of AMPs with conventional antibiotics is also of interest, since it could lead to a reduction in dosage requirements of each agent, treatment duration, and the emergence of resistance [156,160]. This favorable combination could arise not only from the synergy of both antimicrobial activities, but because by acting on the pathogen membranes, AMPs can facilitate the entrance of other drugs. Moreover, due to their immunomodulatory properties, AMPs can be used as adjuvants, priming the immune system in several different manners, such as through the modulation of pro-inflammatory and anti-inflammatory cytokine production, the recruitment, activation, and differentiation of immune cells, the regulation of cell death pathways, and wound healing [152].

Several reports have described the capacity of AMP to kill or inhibit the growth of NTM, including *M. avium*, *M. abscessus*, *M. chelonae*, *M. marinum*, *M. fortuitum*, *M. massiliense*, and *M. kansasii* (Table 3). These peptides come from diverse sources, going from bacteriocins (produced by bacteria) [161,162,163,164,165,166] to mammalian peptides like cathelicidins, human neutrophil peptide (α-defensins), and lactoferricin [167,168,169,170,171], but also including invertebrates like clams and arthropods [172,173,174,175,176]. Nonetheless, as happens for other new treatments, most of these studies are mainly directed towards Mtb, and the experimental use of NTM is often a way to overcome biosafety and experimental problems, or to address the peptides’ activity spectrum. Nonetheless, the increasing number of original reports and reviews describing the antimycobacterial activity of different AMPs highlights the potential of these peptides to be used as new drugs in the fight against mycobacterial infections (Table 3).

### 4.4. Bacteriophages

The emergence of resistance to antibiotics has led to a shift of attention, once again, to the study of bacteriophages as a new strategy with which to combat infections caused by several types of bacteria, including NTM. Bacteriophages are viruses that infect and eventually lyse bacteria [177]. Although still with limitations [178], the use of bacteriophages has several potential advantages, not only as an alternative to conventional antibiotics, but also by working synergistically with them [179]. The fact that they are highly specific on a species or even serovar level makes phages unable to infect host cells, causing no harm to the patient [178,179]. As bacteriophages replicate inside the bacteria, the administered dose can be very low [178]. Besides that, their mode-of-action is much faster than that of antibiotics, and their action is not dependent on the bacterial metabolic state [179].

Bacteriophages are unable to penetrate eukaryotic cell membranes; thus, in the case of intracellular pathogens such as NTM, delivery systems are needed to give them access to the infection sites [178]. Liposomes have been used to carry bacteriophages into infected host cells. As an example, Neith et al. [180] successfully delivered the mycobacteriophage TM4 inside giant liposomes into monocytic THP-1 cells. In many studies, the nonvirulent species of mycobacteria *M. smegmatis* was used as a mycobacteriophage delivery system [181,182,183], since it naturally infects phagocytic cells that host virulent mycobacteria, while allowing, at the same time, the proliferation of the phages inside them [178]. The mycobacteriophage TM4 was effective against *M. avium* and Mtb in vitro [182] and in vivo [183], significantly reducing the number of bacilli. Broxmeyer et al. [182] also proved that the vacuoles containing *M. smegmatis* carrying TM4 fuse with *M. avium*-containing vacuoles in the macrophage, potentiating the effect of the mycobacteriophage. TM4 is only one example among more than 4200 bacteriophages known to infect mycobacteria [178]. More studies need to be carried out with NTM in order to understand which bacteriophages can infect each type of mycobacteria and their mechanism of action.

One way to overcome some of the limitations of bacteriophages, such as the difficulty to find specific phages for a certain type of bacteria, or the development of resistance [178], would be the administration of the bacteriophage-encoded enzymes which are responsible for bacterial lysis, i.e., endolysins. The use of endolysins as enzybiotics, i.e., enzyme-based antibiotics, has been gaining popularity as a new approach with which to combat bacterial antibiotic resistance [179]. The biological strategy of phages to infect bacteria, replicate inside them, and then cause bacterial lysis to release phage progeny, is based in a two-component holin-endolysin system. Holins are responsible for the depolarization of the cytoplasmatic membrane, creating pores that grant endolysins access to the peptidoglycan layer, and then degrading it [179]. Given the high lipidic nature of the mycobacterial cell wall, mycobacteriophages produce a hydrolase, LysA, that targets the peptidoglycan, but also LysB, an esterase that cleaves the linkage between the mycolic acids and arabinogalactan in the mycobacterial outer membrane [184]. Catalão et al. [185] isolated LysA and a shorter protein in the same reading frame from the mycobacteriophage Ms6, both with inhibitory activity against *M. smegmatis*, *M. vaccae*, *M. aurum*, and *M. fortuitum*. Grover et al. [186] reported that LysB isolated from the mycobacteriophage Bxz2 has 10-fold higher esterase activity than LysB isolated from the mycobacteriophage Ms6, but that both effectively inhibit the growth of *M. smegmatis*. Lai et al. [187] showed that LysA and LysB isolated from the mycobacteriophage BTCU-1, besides being active against *M. smegmatis* when administered exogenously, thereby causing severe modifications on the cell wall structure, also inhibited the viability of this mycobacterium growing inside RAW 264.7 macrophages. The promising results of these studies prove that an endolysin approach to treating mycobacterial infections can be applied in the future, also in combination with conventional antibiotics that target the mycobacterial cell wall. Studies specifically directed towards the most clinically-relevant NTM are warranted.

### 4.5. Iron Chelators

Similarly to other pathogenic bacteria, mycobacteria need iron for proliferation and for the establishment of infection [188]. They have actually evolved efficient strategies to acquire iron from the host, such as the synthesis and release of high-affinity siderophores, called mycobactins and carboxymycobactins. These siderophores are able to remove iron from host iron-binding proteins, such as transferrin and lactoferrin [189]. Additionally, *M. tuberculosis* can use haem as an iron source [190,191,192]. Iron is essential, not only because it is an important co-factor in the enzymes involved in bacterial growth, but also because it is needed for some virulence features. For example, the ability of *M. avium* to prevent phagosome maturation inside macrophages was shown to be dependent on its capacity to acquire iron [193]. Iron was also found to be necessary for biofilm formation by *M. smegmatis* [194,195], although this requirement has not been investigated in more clinically-relevant NTM. In agreement with these observations, *M. tuberculosis* with mutations in the proteins involved in iron acquisition was found to exhibit a lower growth in macrophages and also a lower level of virulence in mice [190,196,197].

Given that pathogens fundamentally need iron to survive and proliferate, iron chelators have been suggested as a plausible strategy to treat infections, including mycobacteriosis [198]. We have previously shown that the addition of iron chelators to *M. avium*, in axenic cultures, in macrophage cultures, or in vivo, led to significant decreases in mycobacterial growth [199,200]. Furthermore, we have developed new molecules based on the 3-hydroxy-4-pyridinone iron-chelating moiety, in which the inclusion of a rhodamine residue improved antimycobacterial activity, presumably through improved intracellular distribution and targeting for the mycobacteria-containing phagosome [201,202]. Interestingly, these iron chelators had synergistic activity with ethambutol in decreasing *M. avium* growth inside macrophages [203]. Curiously, in a study aimed at the characterization of the antimicrobial effect of ATP, the authors claimed that ATP inhibits the growth of MAC through iron chelation [204]. In another study, also aimed at the identification of the mechanism of antimycobacterial action of the newly-developed drug PZP, the authors found that it decreases bacterial growth through iron chelation [205].

Given the host’s need for iron for their own metabolic needs, one important concern related to iron chelation therapies is the development of adequate cell-targeting strategies that may guarantee iron depletion in the pathogen without a concomitant deficiency in the host [198]. From the data available so far, chelators alone don’t seem to exhibit a strong enough antimycobacterial activity, but they may have a role as an adjunct therapy, together with conventional antibiotics.

### 4.6. Host-Directed Therapies

Traditional antibiotics act directly on the pathogen by killing or inhibiting its growth. However, by targeting the host factors necessary for the bacteria to survive or replicate, it is possible to control the infection without inducing bacterial resistance and, at the same time, to minimize the dosage of traditional agents [206,207]. Host-directed therapies (HDT) can modulate specific pathways or mechanisms, stimulating, for instance, the host immune system to more efficiently combat the infection, or reducing the symptoms caused by exacerbated inflammation [206,207].

As previously mentioned, an immune response based on IFN-γ producing T cells is very important to control mycobacterial infections. Several studies in mouse models have shown that the manipulation of the immune response may improve infection outcomes. Exogenous administration of recombinant IL-12 enhanced the production of IFN-γ in immunocompetent and immunodeficient SCID or CD4-depleted mice, which conferred protection against some strains of *M. avium* [208]. Also, MAC-infected BALB/c mice injected with IL-18-encoding DNA significantly decreased the bacterial load in the lung through the persistent production of IFN-γ for 8 weeks [209]. On the other hand, the TNF-α inhibitor etanercept, together with conventional antibiotics against TB, significantly decreased the bacterial burden in the lung of Mtb-infected mice, in comparison with treatment with antibiotics alone [210]. There are, however, several studies demonstrating that patients with chronic inflammatory diseases that are treated with TNF-α inhibitors have a higher risk of TB reactivation or NTM disease [211,212]. Besides being one of the major macrophage-activating cytokines [27], TNF-α is essential to maintaining mycobacteria contained in solid, well-defined granulomas in the host lungs, preventing it from infecting other organs [213,214]. However, exaggerated levels of this cytokine are deleterious, as it induces hyperinflammation and the disorganization of granulomas [213]. When considering HDT that interferes in the TNF-α pathway, all these factors must be taken into account.

More recent studies have tried to understand the cause behind attenuated cellular immunity in humans upon infection with mycobacteria. For example, Shu et al. [215] studied 50 patients with MAC lung disease, and compared the response of their peripheral blood mononuclear cells (PBMCs) to MAC antigen stimulation with that of 30 healthy controls. The stimulated PBMCs of patients with MAC lung disease expressed less IFN-γ compared to the healthy controls, but both groups expressed high levels of the receptor programmed cell death-1 (PD-1) and its ligand. PD-1 promotes apoptosis of antigen-specific T cells, attenuating cellular immunity, which allows the infection to progress [216]. By blocking PD-1 and PD ligand with antagonizing antibodies, Shu et al. observed reduced lymphocyte apoptosis and increased production of IFN-γ in cells in both study groups [215]. The authors thus suggest that the PD-1 pathway is an interesting target for HDT, with potential to treat MAC lung disease.

Rather than interfering with systemic immune response, HDT directed to NTM probably relies on the macrophage-intrinsic pathways of bacterial killing. In that direction, an interesting target is autophagy. Autophagy may be triggered by nutrient starvation, when cells reduce mTOR signaling, promoting the digestion of cellular components in autolysosomes [217,218]. But the presence of intracellular pathogens also induces autophagy. This results in the isolation of the pathogen inside autophagosomes, which are fused with lysosomes, resulting in the digestion of the pathogen and the induction of an adaptive immune response [217]. Given the fact that mycobacteria naturally inhibit the phagosome-lysosome fusion [24], the induction of autophagy is an appealing target to HDT [207]. Studies done with nonpathogenic mycobacteria have yielded contradictory results, and it is not clear whether mycobacteria induce or inhibit macrophage autophagy, and what the role of this process is in mycobacterial growth containment. [217,219]. Anticonvulsant drugs, EGFR inhibitors, and even the role of vitamin D in promoting autophagy have been studied against Mtb [207]. In *M. avium*, lactoferricin, in particular, the D-enantiomer of LFcin17-30, leads to the intramacrophagic death of the mycobacteria by inducing phagosomal maturation and autophagy, with increased lysosomes and autophagosomes [171]. More studies must be performed to find an HDT target, perhaps in the autophagy pathway, which can overcome the mechanism of the arrest of phagosome maturation by NTM.

## 5. Concluding Remarks

Infectious diseases continue to pose a heavy burden on public health systems worldwide. NTM were, for a long time, considered nonpathogenic or rare opportunistic agents. However, the incidence of disease by these agents is rising, and is revealing the medical community to be devoid of validated, efficient therapeutic weapons to fight them. With this review, we hope to stimulate researchers in the area of drug development to test atypical therapeutic agents against NTM, in an effort to put forward new, efficient alternatives with which to treat these devastating diseases.

## Figures and Tables

**Figure 1 antibiotics-09-00018-f001:**
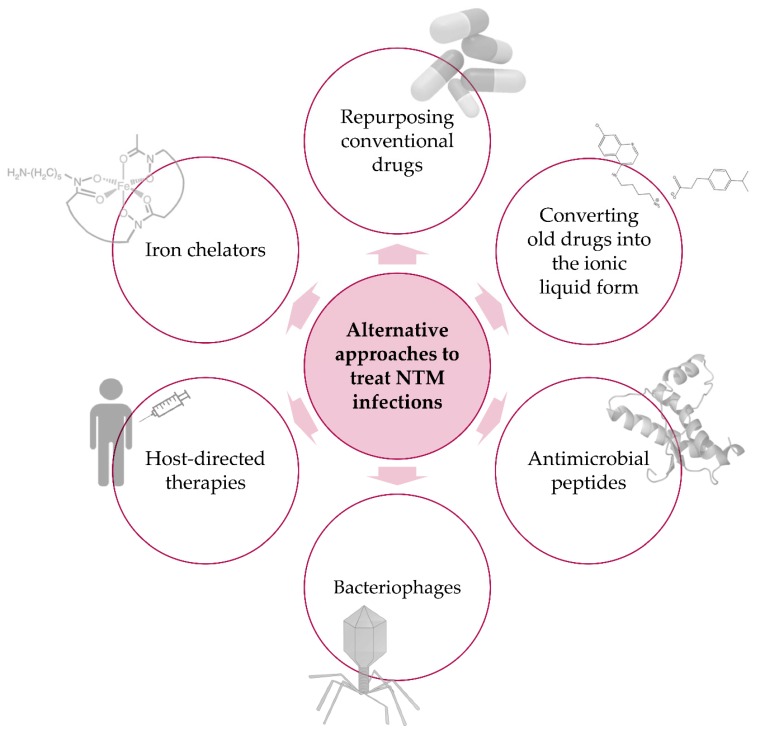
Alternative approaches to treat NTM infections. Schematic representation of new, promising alternatives to combat infectious diseases caused by NTM species.

**Table 1 antibiotics-09-00018-t001:** Summary of recent publications showing new, promising antimicrobial agents against NTM species.

Compound	Description	NTM	Ref.
Nitric oxide	Inhaled nitric oxide to treat MABC lung disease with ongoing clinical trials. Shows synergistic effect with antimycobacterial antibiotics, such as clofazimine.	*M. abscessus*	[109,110,111]
PIPD1	Piperidinol-based molecule that targets mycolic acid transport.	*M. abscessus*	[112]
Indolecarboxamide analogs	Structure-activity relationship studies of a series of indolecarboxamide analogs that target mycolic acid transport.	*M. abscessus*; *M. chelonae*; *M. massiliense*; *M. bolletii*; MAC; *M. xenopi*	[113,114]
Benzimidazole SPR719	Active form of the prodrug SPR720, is an aminobenzimidazole that inhibits the ATPase activity of gyrase in Mtb.	MAC; MABC; *M. chelonae*; *M. immunogenum*; *M. fortuitum*; *M. mucogenicum*; *M. kansasii*; *M. marinum*; *M. simiae*	[115,116]
TP-271	Novel fluorocycline antimicrobial related to tetracycline; active in vitro against NTM isolates.	*M. abscessus*; *M. fortuitum*	[117]
CyCs	Cyclipostins and cyclophostin analogs with selective in vitro and intramacrophagic activity against mycobacteria; mechanism of action related to enzyme-inhibition involved in lipid metabolism and/or cell wall biosynthesis.	*M. abscessus*; *M. marinum*; *M. smegmatis*	[118,119]
Salicylanilide esters, carbamates and benzoates	De novo synthesized molecules with in vitro potency against *M. abscessus*; ability to inhibit various bacterial enzymes and to function as proton shuttles, destroying the cellular proton gradient killing the bacteria.	*M. abscessus*; *M. avium*; *M. kansasii*	[120,121]
Capuramycin analogs	Nucleoside antibiotics that target peptidoglycan synthesis, with in vitro activity against several species of NTM.	MAC; *M. paratuberculosis*; *M. kansasii*; *M. abscessus*; *M. smegmatis*; *M. ulcerans*	[122,123]
ACH-702	Isothiazoloquinolones, analogs related to quinolones, which target bacterial replication; in vitro activity against NTM.	MAC; *M. fortuitum*	[124]
IAPs	Imidazo [1,2-a]pyridine-3-carboxamides; potential in vitro and in vivo activity against MAC.	MAC	[125]

**Table 2 antibiotics-09-00018-t002:** Summary of recent publications showing repurposed drugs with favorable activity against NTM infections.

Compound	Description	NTM	Ref.
Carvacrol	Major constituent of many essential oils of the Labiatae family; Generally recognized as safe (GRAS) and approved for use in food; Antioxidant, anti-inflammatory, antitumor, analgesic, antihepatotoxic, and insecticidal activities; Activity in vitro against planktonic and biofilm cells of several RGM.	*M. abscessus*; *M. fortuitum*; *M. chelonae*; *M. mucogenicum*; *M. smegmatis.* Biofilm inhibiting activity	[128,129]
Omadacycline	Tetracycline, used for skin infections and community-acquired pneumonia caused by Gram-positive bacteria; In vitro activity against *M. abscessus*.	*M. abscessus*; *M. chelonae*; *M. fortuitum*	[130,131,132,133]
Mefloquine and enantiomers	Derivative of 4-quinolinemethanol; An antimicrobial drug used against chloroquine-resistant *Plasmodium falciparum*; Active in vitro and in vivo against MAC; Synergistic effect with antimycobacterial drugs in vivo.	MAC	[134,135,136]
Thioridazine	Phenothiazine derivative, an antipsychotic drug with activity against Mtb, by inhibition of the electron transport chain; In vitro activity in a hollow-fiber system model for pulmonary MAC disease (HFS-MAC).	MAC	[137,138]
Chloroquine	Antimalarial with activity in vitro and in vivo against *M. avium*. Also active in vitro against HIV-1.	MAC	[139]
Primaquine	Urea derivatives of this antimalarial showed high activity in vitro against *M. avium*.	MAC	[140]

**Table 3 antibiotics-09-00018-t003:** Summary of AMP activity against NTM.

AMP	Origin	NTM Species	Activity	Ref.
Ecumicin	Extracts from actinomycetes	*M. abscessus*; *M. chelonae*; *M. marinum*; *M. kansasii*; *M. avium*	Axenic	[162]
Lassomycin	Extracts from actinomycetes	*M. avium*	Axenic	[163]
Nisin	*Lactococcus lactis*	*M. paratuberculosis*	Axenic	[166]
Nisin A, S, T, and V	*Lactococcus lactis*	*M. kansasii*; *M. avium*	Axenic	[165]
Lacticin 3147	*Lactococcus lactis*	*M. kansasii*; *M. avium*	Axenic	[164]
LL-37	Human Cathelicidin	*M. avium*	Macrophages	[167]
LLKKK-18 (plus nanoparticles)	Cathelicidin LL-37	*M. marinum*	Axenic; macrophages	[168]
NK-2 (plus nanoparticles)	NK cells and cytotoxic T cells	*M. marinum*	Axenic; macrophages	[168]
HNP-1, 2 and 3	Human neutrophils	*M. avium*	Axenic	[169]
hLFcin1-11 and variants	Human lactoferricin	*M. avium*	Axenic	[170]
LFcin17-30 and variants	Bovine lactoferricin	*M. avium*	Axenic; macrophages	[170,171]
Mcdef	Manila clams (*Ruditapes philippinarum*)	*M. fortuitum*	Axenic	[172]
NDBP-5.5	Scorpion (*Hadrurus gertschi*)	*M. abscessus*	Anexic; macrophages; in vivo	[173]
ToAP2	Scorpion (*Tityus obscurus*)	*M. massiliense*	Axenic; macrophages; in vivo	[175]
Polydim-I	Wasp (*Polybia dimorpha*)	*M. abscessus*	Anexic; macrophages; in vivo	[174]
Polybia-MPII	Mastoparans from wasp (*Pseudopolybia vespiceps*)	*M. abscessus* sp. *massiliense*	Axenic; macrophages	[176]

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
