# Peer review of "Looking beyond Typical Treatments for Atypical Mycobacteria"

_antibiotics, 2020, doi:10.3390/antibiotics9010018_

Round 1

Reviewer 1 Report

The manuscript by Bento et al entitled "Looking beyond Typical Treatments for Atypical Mycobacteria" is a review of recent findings in the area of treatments for slow and rapidly growing non-tuberculosis mycobacteria. The authors cover the epidemiology and clinical presentation of NTM and then proceed to discuss various current treatment options, including first and second line treatments. They then assess areas in which further work may yield the development of new treatments , including repurposing of drugs, bacteriophages, antimicrobial peptides, etc.

The review is in general well written and the topics are well covered, with each area sufficiently referenced. Overall, I think this is a well written review of this important area that often does not receive a lot of coverage.

I have only some minor comments:

The manuscript focuses primarily of pulmonary non-tuberculosis mycobacterial pathogens (M. abcessus, etc). Technically, organisms such as M. marinum and M. ulcerans, which do not cause pulmonary disease, are also non-tuberculosis mycobacteria. I'm not suggesting these other organisms should be included in the review, but perhaps the authors could make some statement within the introductory paragraphs that their review will primarily focus on pulmonary NTM pathogens and their treatment. There are some minor unusual English expressions and grammatical errors within the manuscript, but these do not generally affect the ability of the reader to understand the points made Minor expression errors: L54 - "high resistance to environmental aggressions and…". "Harsh environments" would be better than "environmental aggressions" L96 "individuals that are fragilized by". Replace "fragilized" by "sensitized". L308 "many researchers turned to the screen of hundreds". Replace "turned to the screen" with "turned to screen". L484 "host’s need for iron for his own metabolic needs…". Gender neutral is best here - replace "his" with "their".

Author Response

Thank you very much for your comments.

We completely agree that organisms such as M. marinum and M. ulcerans are also non-tuberculosis mycobacteria with an important impact in human disease, but in fact the treatment of these types of infections was beyond the scope of our review. To make it more clear that this manuscript focuses on pulmonary infections, we added one sentence in the introduction (lines 37-39).

We made the alterations suggested on lines 54 (now 56), 96 (now 98), 308 (now 310) and 484 (486).

Best regards.

Reviewer 2 Report

In this review manuscript, entitled “Looking beyond typical treatments for atypical Mycobacteria,” the authors are trying to describe nontuberculous mycobacteria (NTM) and diseases associated with them, identify limitations of new treatments and alternative approaches against NTM.

The primary strengths of this manuscript are its careful evaluation of the limitations of new treatments against NTM, different alternative approaches with an emphasis on antimicrobial peptides, bacteriophages, iron chelators, and hots-immune based therapies. In general, the manuscript is well-written; however, this manuscript has a few minor issues that should be addressed.

Lines 144-146> It induces……………. mycobacterial infections [27], are less clear, needs to be rephrased. CD4+T cells, + should be superscripted (+) Line 306>The word “in” should be replaced by “is”.

Author Response

Thank you very much for the comments.

We have re-phrased lines 140-146, according to the suggestions. We also corrected the typo on line 306.